# Hemodynamic molecular imaging of tumor-associated enzyme activity in the living brain

Mitul Desai[1], Jitendra Sharma[2], Adrian L Slusarczyk[1], Ashley A Chapin[1], Robert Ohlendorf[1], Agata Wisniowska[3], Mriganka Sur[2], Alan Jasanoff[1,2,4]*

[1]Department of Biological Engineering, Massachusetts Institute of Technology, Cambridge, United States; [2]Department of Brain & Cognitive Sciences, Massachusetts Institute of Technology, Cambridge, United States; [3]Harvard-MIT Health Sciences and Technology, Massachusetts Institute of Technology, Cambridge, United States; [4]Department of Nuclear Science & Engineering, Massachusetts Institute of Technology, Cambridge, United States

**Abstract** Molecular imaging could have great utility for detecting, classifying, and guiding treatment of brain disorders, but existing probes offer limited capability for assessing relevant physiological parameters. Here, we describe a potent approach for noninvasive mapping of cancer-associated enzyme activity using a molecular sensor that acts on the vasculature, providing a diagnostic readout via local changes in hemodynamic image contrast. The sensor is targeted at the fibroblast activation protein (FAP), an extracellular dipeptidase and clinically relevant biomarker of brain tumor biology. Optimal FAP sensor variants were identified by screening a series of prototypes for responsiveness in a cell-based bioassay. The best variant was then applied for quantitative neuroimaging of FAP activity in rats, where it reveals nanomolar-scale FAP expression by xenografted cells. The activated probe also induces robust hemodynamic contrast in nonhuman primate brain. This work thus demonstrates a potentially translatable strategy for ultrasensitive functional imaging of molecular targets in neuromedicine.

*For correspondence: jasanoff@mit.edu

Competing interest: The authors declare that no competing interests exist.

## Editor's evaluation

Molecular probes that respond to disease-specific activities to produce a diagnostic readout have had a major impact in the clinical management of cancer. The current study describes a genetically engineered sensor for the cancer-associated, fibroblast activation protein, which reports via local changes in hemodynamic image contrast using MRI. Development of activity based imaging probes is an important area of study for advancing precision medicine and may more accurately represent disease prognosis and stratification over conventional imaging probes.

## Introduction

Detection of molecular epitopes is vital to effective classification of cancers and determination of therapeutic strategies (*Olar and Aldape, 2014*; *Reifenberger et al., 2017*; *Wen and Reardon, 2016*). In the brain, performing such diagnoses is particularly complicated by the difficulty of obtaining biopsy samples for ex vivo analysis. The development of sensitive in vivo assays for tumor-associated proteins is therefore an important goal in brain cancer research and treatment. Extracellular proteases have emerged as notable hallmarks of pernicious cancers such as gliomas (*Cohen and Colman, 2015*; *Edwards et al., 2008*). Several proteases are highly expressed on the surfaces of glioma cells and

**Figure 1.** Design of fibroblast activation protein (FAP)-sensitive vasoactive imaging probes. (**A**) FAP-sensitive vasoprobes are constructed by fusing the vasoactive peptide calcitonin gene-related peptide (CGRP) (cyan) via a FAP-cleavable linker to a steric blocking domain (green) that prevents CGRP from acting on its receptor when present. Cleavage by cell surface-expressed FAP (red) removes the blocking domain and allows the CGRP domain to stimulate the RAMP/CLR receptor on vascular smooth muscle cells, promoting intracellular cAMP accumulation, vasodilation, and production of consequent hemodynamic imaging signals (right). (**B**) Candidate FAP sensors designed with biotin as a blocking domain and FAP-sensitive linkers for constructs (1–5) as shown. (**C**) Luminescence readouts obtained using a cell-based RAMP/CLR activation bioassay during stimulation with varying concentrations of construct (5), with (red) or without (gray) pretreatment by 10 ng/µL FAP. Data points reflect mean and SD (error bars) of three technical replicates each. (**D**) Midpoints for receptor activation ($EC_{50}$ values) measured using the bioassay of panel (**C**) for constructs (1–5) and control peptide (CGRP) with (red) or without (gray) FAP pretreatment.

The online version of this article includes the following figure supplement(s) for figure 1:

**Figure supplement 1.** Assessment of candidate fibroblast activation protein (FAP) sensors in vitro.

play key roles in infiltration of tumors into brain tissue (*Mentlein et al., 2012*). These enzymes are thus attractive both as diagnostic markers and as targets for therapy (*Vandooren et al., 2016*; *Verdoes and Verhelst, 2016*). Because the profiles of protease expression differ among disease subtypes, detecting proteases on cancer cells in situ could be of critical importance in classifying and subsequently treating brain tumors.

Noninvasive measurement of tumor-associated protease activity inside the living brain is exceedingly challenging, but we recently introduced a family of ultrasensitive molecular probes with the potential to address this need (*Desai et al., 2016*). The probes are derived from peptides that induce vasodilation and consequent hemodynamic contrast detectable by many brain imaging modalities, most prominently including magnetic resonance imaging (MRI). Protease sensors can be engineered by fusing such peptides to protease-labile blocking domains; enzymatic cleavage of the blocking moieties activates the sensors, creating an imaging signal (*Figure 1A*). This principle was illustrated in proof-of-concept experiments that employed vasoactive probes (vasoprobes) based on the calcitonin gene-related peptide (CGRP), a 37-residue molecule that induces dilation of intracerebral arterioles with a 50% effective concentration ($EC_{50}$) below 10 nM (*Chin et al., 1994*; *McCulloch et al., 1986*), approaching the low doses of radiotracers used in nuclear imaging. Importantly, CGRP is a potent endogenous vasodilator in humans, raising prospects that CGRP-based vasoprobes could be applied as non-immunogenic imaging agents in the clinic (*Brain et al., 1985*; *Howden et al., 1988*; *Li et al., 2007*; *Warren et al., 1992*).

In this work, we introduce a vasoprobe-based cancer imaging strategy that targets the fibroblast-associated protein (FAP), an extracellular protease that is highly expressed in some glioblastomas (*Busek et al., 2008*), as well as tumorigenic stromal tissues (*Puré and Blomberg, 2018*). In tissue isolates from patients, FAP is most highly expressed in mesenchymal cells and in grade IV tumors, which demand therapeutic interventions distinct from other glioblastoma subtypes (*Busek et al.,*

2016). FAP expression levels also differ among established glioma cell lines (*Busek et al., 2008*), indicating the utility of FAP measurements as a basis for characterizing brain tumors. We show that vasoprobes enable wide-field molecular imaging of FAP activity using a contrast mechanism that is effective in rodent and primate brains, thus demonstrating a powerful avenue for in situ evaluation of neurological markers in living subjects.

## Results

### Molecular engineering of a FAP-responsive vasoprobe

To identify FAP-sensitive vasoprobes, we began by creating a series of sterically inhibited CGRP derivatives that we predicted could undergo FAP-dependent activation. In light of previous evidence that N-terminal extension of CGRP sharply reduces receptor-binding activity (*Desai et al., 2016*), we formed the candidate sensors by adding biotinylated FAP-cleavable domains to the N-terminus of human α-CGRP sequence (*Figure 1B* and *Supplementary file 1*). FAP-labile segments consisted of glycyl-proline consensus sites (*Aertgeerts et al., 2005*) flanked by polypeptide segments of varying length and composition. Each of these molecules was evaluated in a cell-based bioassay for CGRP receptor agonism (*Desai et al., 2016*; *Figure 1C*, *Figure 1—figure supplement 1*). Data from this bioassay provide a basis for estimating $EC_{50}$ values for receptor activation that can then be compared across conditions.

Prior to FAP digestion, the vasoprobes display $EC_{50}$ values ranging from 6.7 to 21 nM, corresponding to reductions of 81- to 260-fold in potency with respect to unmodified CGRP (*Figure 1D*). Following treatment with FAP, the activity of each blocked vasoprobe is increased to levels approaching that of native CGRP, however. The greatest FAP-mediated change in CGRP receptor agonism is observed for construct (5), which contains two tandem cleavage sites and exhibits $EC_{50}$ values of 6.7 ± 0.2 nM and 0.20 ± 0.01 nM before and after FAP processing, respectively—a 34-fold increase in activity. We henceforth refer to this molecule simply as the FAP-sensitive vasoprobe (FAPVap).

### FAPVap-based molecular imaging in vivo

To examine the ability of FAPVap to measure FAP expression on cancer cells in the living brain, we tested the probe on a xenograft model. HEK293 cell lines were generated by stable transfection with constructs encoding human FAP with the fluorescent marker mKate2 (test) or mKate2 only (control) and assessed for enzymatic activity in cell culture (*Figure 2—figure supplement 1* and *Supplementary file 2*). Measurements calibrated with known amounts of FAP indicated activity equivalent to 8.5 ± 4.6 nM soluble enzyme (mean ± SD, n = 3), corresponding to an estimated mean of 8.5 pmol FAP per $10^6$ test cells. Approximately $3 \times 10^5$ test or control cells were implanted at bilaterally symmetric sites in rat striatum and adapted to their environment over 1 day prior to imaging experiments (*Figure 3—figure supplement 2*). To assess FAPVap activation by the cells after the adaptation period, 100 nM of the probe was perfused for 10 min over each xenograft during continuous $T_2$-weighted MRI scanning. Data indicate clear enhancement of imaging signal in the neighborhood of FAP-expressing but not control cells (*Figure 2A*), consistent with the expected difference between activated and inactivated FAPVap. MRI signal increases begin during FAPVap infusion and rise toward an asymptote reached ~10 min after infusion offset (*Figure 2B*). By the end of this trajectory, mean signal changes of 13.1% ± 1.9% and 0.7% ± 1.5% were observed in the presence of test and control xenografts, respectively, a significant difference (t-test p=0.007, n = 3) that indicates that FAP-expressing cancer cells are selectively and robustly detected by the FAPVap sensor in vivo (*Figure 2C*).

To analyze MRI signal change dynamics in more mechanistic terms, we postulated a kinetic model that accounts for FAPVap infusion, enzymatic conversion of the probe to the CGRP-based product (CGRP*), consequent hemodynamic responses, and probe removal (*Figure 2D*). For the sake of simplicity, the model assumes that hemodynamic signal changes are directly proportional to the activated probe concentration and that enzymatic processing obeys first-order kinetics, as expected from published FAP kinetic constants (*Edosada et al., 2006*) relevant to FAPVap concentrations used in our experiments. A key parameter in this model is the effective first-order rate constant for product formation from FAPVap, $k$. This rate constant is approximately equal to $k_{cat}[FAP]/K_m$ in the Michaelis–Menten formalism, where $k_{cat}$ and $K_m$ are the turnover rate and Michaelis constants, [FAP] is the local enzyme concentration, and the FAPVap concentration is much lower than $K_m$, as we expect in our

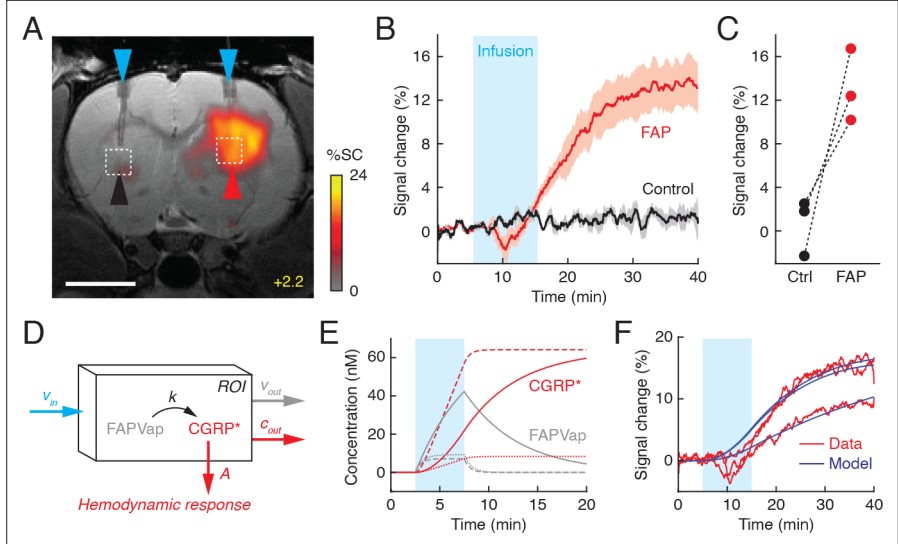

**Figure 2.** Fibroblast activation protein-sensitive vasoprobe (FAPVap) reports FAP enzyme activity in vivo. (**A**) MRI data from an individual rat implanted with FAP-expressing or control tumor cells (red and black arrowheads, respectively) in the dorsal striatum at bregma +2.2 mm. 100 nM FAPVap was infused over the tumors via cannulae (blue arrowheads), resulting in signal changes observed in the presence but not the absence of FAP expression. Color scale denotes percent $T_2$-weighted MRI signal changes (%SC) recorded after infusion with respect to the pre-infusion baseline. Grayscale image underlay is an anatomical scan. Scale bar = 4 mm. (**B**) Mean time courses of MRI signal expressed as percent change observed before, during (cyan box), and after FAPVap infusion from regions of interest (ROIs) defined with respect to FAP-expressing (red) or control (black) xenograft locations in three animals (*cf.* dashed boxes in **A**). Shading denotes SEM (n = 3). (**C**) Magnitudes of FAPVap responses observed in paired probe injections over control (Ctrl) and FAP-expressing tumors. (**D**) Diagram of a kinetic model accounting for rates of FAPVap injection ($v_{in}$), FAP-catalyzed conversion from FAPVap to CGRP* ($k$), and efflux of FAPVap and CGRP* ($v_{out}$ and $c_{out}$) from an ROI. Hemodynamic responses are related to CGRP* concentration by a constant of proportionality, $A$. (**E**) Concentration dynamics of FAPVap (gray) and CGRP* (red) simulated using the model of (**D**). Calculations assume that 100 nM FAPVap is infused into an ROI containing 1 nM (solid lines) or 10 nM FAP (dashed lines) with $v_{out} = 0$, or containing 1 nM FAP with $v_{out} = 0.01$ s$^{-1}$ (dotted lines). See text for further details. (**F**) Fitting of model curves (blue) to ROI-averaged FAPVap imaging time courses (red) from three animals, showing that broad features of the MRI signal are closely approximated, with the exception of suspected pressure artifacts during the infusion period.

The online version of this article includes the following figure supplement(s) for figure 2:

**Figure supplement 1.** Fibroblast activation protein (FAP) expression in transfected cells.

**Figure supplement 2.** Microscopic visualization of brain xenografts.

experiments. Realistic parameter choices allow our model to approximate the observed time course of MRI signal change in *Figure 2B*, including the slow initiation of MRI responses and protracted increase in image signal that extends past the FAPVap infusion period (*Figure 2E*). This behavior is observed under conditions where the efflux rate is zero for both FAPVap and its CGRP* product. Variation of parameter choices alters the shape of the profile, however; for instance, modeling probe dynamics with 10-fold greater FAP concentration or inclusion of a moderate FAPVap removal rate constant (0.01 s$^{-1}$) leads to a roughly linear increase in CGRP* concentration during infusion, followed by a flat asymptote.

Fitting of the resulting model to actual data enables estimation of $k$ values from our imaging data (*Figure 2F*). Calculations assume a probe infusion rate of 0.11 nM/s, computed based on the infusion rate, FAPVap concentration, and region of interest (ROI) volume used in *Figure 2B and C*. The three time courses corresponding to *Figure 2C* were optimally approximated using $k$ values ranging from $1.2 \times 10^{-4}$ to $6.8 \times 10^{-4}$/s, hemodynamic response coefficients of 0.5–1.0%/nM, and rate constants for FAPVap and CGRP* removal all below 10$^{-3}$ s$^{-1}$. Fitted curves do not emulate the dip in MRI signal observed for some traces during the FAPVap injection period, which we take to be an infusion-related

artifact. Despite the assumptions involved, these results establish the feasibility of model-based interpretation of vasoactive molecular imaging data in pseudo-quantitative terms.

## Wide-field measurement of tumor-associated FAP activity

In a clinical context, unbiased detection of brain tumor markers should be performed using noninvasive delivery modes that do not require prior localization of or intracranial access to tumor sites. For imaging agents that are not known to permeate the blood–brain barrier, this can be accomplished using intrathecal infusion of probes into the cerebrospinal fluid (CSF). To emulate this scenario in the rat, we assessed the ability of FAPVap to report the location and activity of FAP-expressing cell implants following distal infusion of the probe into a CSF reservoir remote from xenograft locations in the brain (**Ohlendorf et al., 2020**). 100 nM of the probe was injected at a rate of 1 µL/min into the

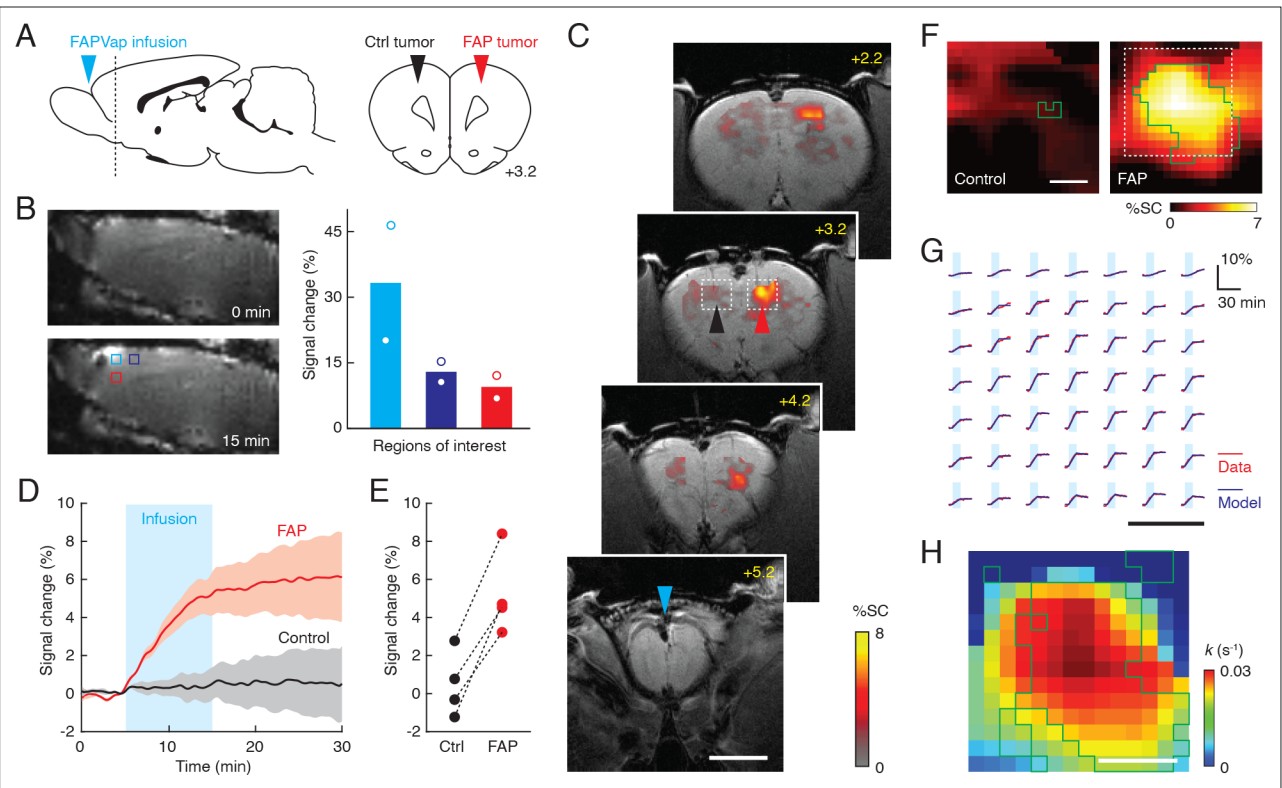

**Figure 3.** Wide-field imaging of fibroblast activation protein (FAP)-expressing tumors in vivo. (**A**) Geometry employed for wide-field molecular imaging of FAP activity in rat brain tumors. Blue arrowhead denotes probe infusion site. Horizontal dashed line in sagittal view (left) denotes position of coronal section (right) showing location of implanted FAP-expressing and control tumors with respect to brain atlas sections (**Paxinos and Watson, 2009**). (**B**) Infusion of gadolinium-diethylenetriamine pentaacetic acid (Gd-DTPA) via the intra-cerebrospinal fluid (intra-CSF) route diagrammed in (**A**) results in broad signal enhancements when comparing $T_1$-weighted MRI scans obtained before (top left) and after (bottom left) contrast agent infusion. Plot at right shows signal changes observed in the three color-coded regions of interest (ROIs) shown at bottom left, indicating inverse dependence of signal enhancements on distance from the infusion site (n = 2). (**C**) Coronal scans of a rat showing infusion location (bregma +5.2 mm) and profiles of $T_2$*-weighted MRI signal change (color) following FAP-sensitive vasoprobe (FAPVap) infusion in the presence of control (black arrowhead) and FAP-expressing (red arrowhead) xenografts. The gray underlay is an anatomical scan and the scale bar = 4 mm. (**D**) Mean signal change time courses with respect to baseline averaged over ROIs defined with respect to xenografts in four animals. Cyan box denotes the FAPVap infusion period, and shaded intervals denote SEM (n = 4). (**E**) FAPVap response amplitudes corresponding to the 15 min time point in (**D**). (**F**) Maps of mean MRI signal changes observed near FAP-expressing and control tumors (n = 4 each), corresponding to boxed regions as in panel (**C**). Green outlines denote voxels with responses significantly different from zero (t-test p≤0.05, n = 4). Scale bar = 0.5 mm. (**G**) Analysis of time courses in the boxed region of (**F**) using the kinetic model of **Figure 2D**, showing comparison of fitted models (blue) to data (red) in a regular grid of individual voxels evenly spaced over the dashed box in panel (**F**). Scale bar = 0.5 mm. (**H**) Values of the FAP catalysis rate k obtained by fitting the FAPVap kinetic model to the data in the dashed box of panel (**F**). Green outlines denote values with t-test p≤0.05 (n ≥ 2). Scale bar = 0.5 mm.

The online version of this article includes the following figure supplement(s) for figure 3:

**Figure supplement 1.** Wide-field molecular imaging of fibroblast activation protein (FAP) activity in rat brain.

**Figure supplement 2.** Fibroblast activation protein (FAP) activity detection sensitivity.

subarachnoid space caudal to the olfactory bulb but rostral to test and control tumor cell implantation sites in dorsal cortex (*Figure 3A*). Application of this procedure to infusion of the $T_1$-weighted MRI contrast agent gadolinium-diethylenetriamine pentaacetic acid (Gd-DTPA) results in broad signal enhancements across the rostral brain, showing that vasoprobes infused via this route should be able to access the xenografts (*Figure 3B*).

Hemodynamic molecular imaging of the entire rostral brain during remote intra-CSF infusion of FAPVap was performed using a more sensitive $T_2$*-weighted MRI acquisition approach than the $T_2$-weighted scheme of *Figure 2*. Results reveal the induction of signal changes in the neighborhood of the FAP-expressing cells but not control cells (*Figure 3C*, *Figure 3—figure supplement 1*). Image signal observed near FAP xenografts began increasing upon onset of probe infusion, rose linearly during the injection period, and reached an asymptote at the end of the infusion period (*Figure 3D*), more closely paralleling the dotted or dashed model curves of *Figure 2E* than the experimental data of *Figure 2B*. Immediately following infusion, mean MRI signal changes observed near FAP and control tumors were 5.2% ± 1.1% and 0.5% ± 0.9%, respectively (*Figure 3E*), a significant difference (*t*-test p=0.015, n = 4). The average spatial profile of hemodynamic responses around FAP-expressing xenografts was determined by coregistering xenograft sites from four independent experiments. Mean FAPVap-mediated responses are shown for test and control xenografts in *Figure 3F* and indicate the presence of consistent responses across a roughly 1 mm diameter region around FAP-expressing cells.

Probe responses within regions that underwent signal changes could be fit at a single-voxel level to the model of *Figure 2D*. Using a probe infusion rate of 0.011 nM/s estimated by assuming inverse-squared dependence on distance from the CSF infusion site, a hemodynamic amplitude of 20% per nM activated FAPVap produced an optimal fit to the data over the region analyzed. Across this region, signal changes within 95% of the individual voxels could be fit to the model with $R^2$ values over 0.9 (*Figure 3G*). FAP-catalyzed cleavage rate constants estimated using this approach fall in the range from $k$ = 0–0.03 $s^{-1}$, with probe efflux rates averaging ~0.4 $s^{-1}$, but CGRP* efflux rates below $10^{-3}$ $s^{-1}$. Consistency of FAPVap cleavage rate constants across animals is reflected by statistical significance (*t*-test p≤0.05, n ≥ 2) of nonzero values in over 50% of voxels (*Figure 3H*). Discrepancies in these fitted parameters with respect to the results of *Figure 2E* may reflect a combination of differences in MRI acquisition protocol, probe delivery method, and xenografts themselves. The analysis presented here nevertheless describes a spatial profile of relative FAP-dependent cleavage rate constants, where higher $k$ values observed near the center of the field of view presumably reflect a greater effective concentration of FAP itself in these regions. Analysis of signal-to-noise and comparison of signal change distributions with fitted $k$ values suggests that activity levels of $k \geq 0.019$ $s^{-1}$, corresponding to estimated concentrations of about 13 nM FAP or more, should be detectable at a single voxel and single animal level using our current methods (*Figure 3—figure supplement 2*); yet greater sensitivity could be achievable at an ROI level or across multiple experiments, or using different imaging protocols. These results thus collectively demonstrate that wide-field FAPVap-mediated tumor model imaging enables quantitative estimation of FAP enzymatic activity levels in vivo.

## Clinical applicability of FAPVap-based imaging

Translational applications of FAPVap would require that the probe be able to sense physiological levels of FAP found on patient-derived glioma cells and that the vasoprobe-based contrast mechanism operates robustly in human subjects. To compare the HEK cell-based model we investigated in *Figure 3* with more clinically relevant specimens, we compared FAPVap cleavage by the FAP-expressing cells used in our in vivo experiments with cleavage by the established U138 malignant glioma cell line, which naturally expresses FAP at substantial levels (*Busek et al., 2008*). Measurements using the bioassay of *Figure 1* show that both the transgenic and glioma cells perform similarly (*Figure 4A*), with significant FAP activity levels detected in both cases for $10^4$ cells or more (*t*-test p≤0.002, n = 3). Quantification of the bioassay results with respect to known concentrations of FAP under identical conditions (*Figure 4B*) indicates that levels of approximately 1 pg/cell FAP are present in both samples, consistent with the fluorescent assay of *Figure 2—figure supplement 1*. Comparison of U138 FAP expression with literature reports indicates that several glioma cell lines derived from human tumors express higher FAP levels (e.g., U118, U343), while some others express less FAP (e.g., U87, U373) (*Busek et al., 2008*; *Mentlein et al., 2011*). This indicates that FAP expression by

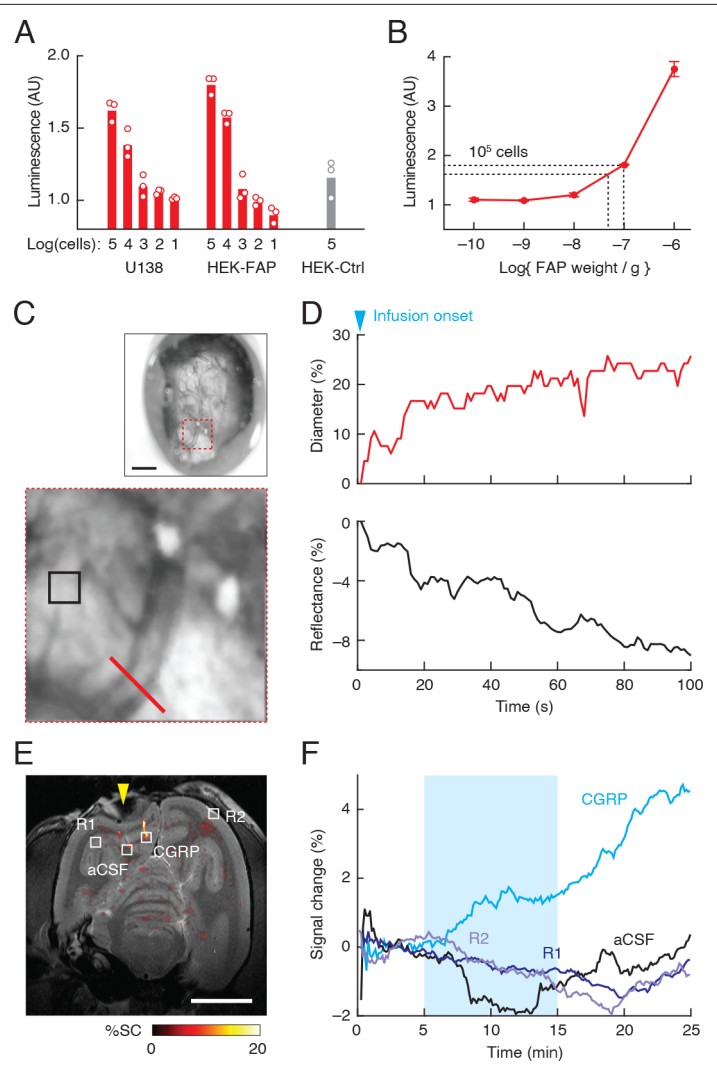

**Figure 4.** Clinical applicability of fibroblast activation protein-sensitive vasoprobe (FAPVap)-based imaging. (**A**) Luminescence response measured after addition of FAPVap to varying numbers of U138 human glioma cells, FAP-transfected HEK cells (HEK-FAP) and control untransfected HEK cells (HEK-Ctrl), using the bioassay of *Figure 1*. Values normalized to the response to uncleaved FAPVap without cells. (**B**) FAPVap processing by known amounts of FAP under the same assay conditions as (**A**). Dashed lines indicate mean assay results obtained from $10^5$ U138 and HEK-FAP cells, indicating the presence of roughly 1 pg FAP/cell, according to the standard curve. (**C**) Optical imaging of responses to vasoprobe infusion into a cranial chamber (top, *Figure 4—figure supplement 1*) implanted over the cortex of a squirrel monkey. Scale bar = 1 mm. Close-up (bottom) corresponds to red dashed square at top and shows a single blood vessel (red bar), as well as reflective parenchymal tissue (black square). (**D**) Top: diameter of the vessel labeled in (**C**) over 100 s immediately following onset of perfusion with 100 nM calcitonin gene-related peptide (CGRP), equivalent to cleaved FAPVap. Bottom: reflectance from the parenchymal region of interest (ROI) labeled in (**E**) over the same time period. (**E**) Map of MRI signal changes elicited by injection of CGRP or artificial cerebrospinal fluid (aCSF) into the squirrel monkey brain. R1 and R2 denote two uninjected control regions. Yellow arrowhead denotes the position of the cranial chamber in (**C**). Scale bar = 1 cm. (**F**) Time courses of MRI signal change observed in the four regions designated in (**E**), before, during (cyan shading), and after vasoprobe infusion.

The online version of this article includes the following figure supplement(s) for figure 4:

**Figure supplement 1.** Optical window implant for cortical surface imaging in monkeys.

glioblastoma cells in vivo is likely to be comparable to the cellular model we probed in *Figure 3*, and therefore that the FAPVap sensor could be appropriate for distinguishing or staging analogous human cancers.

To further investigate the feasibility of vasoprobe-based diagnosis in human brains, we applied α-CGRP—which closely emulates enzyme-activated FAPVap—in a nonhuman primate and evaluated its ability to induce vasodilation and hemodynamic MRI contrast. A squirrel monkey was implanted with a cranial window that permits probe perfusion over the exposed cortex during optical imaging (*Figure 4C*, *Figure 4—figure supplement 1*). Introduction of 100 nM vasoprobe into the perfusion chamber induced a rapid decrease in cortical reflectance, as well as increases in the diameters of observable blood vessels (*Figure 4D*). These results are consistent with probe-mediated increases in capillary and arterial blood flow and align well with previously reported response characteristics in rodents (*Desai et al., 2016*).

To assess vasoprobe-mediated MRI contrast changes, test and control injections were performed into the cortical parenchyma during serial acquisition of $T_2$*-weighted scans (*Figure 4E*). Infusion of 100 nM CGRP produced a regional signal increase of ~5% that was not observed following control infusion of artificial cerebrospinal fluid (aCSF) or in reference brain regions that did not receive injections (*Figure 4F*). The results of this test therefore confirm that the FAPVap contrast mechanism operates in primates and indicate the potential suitability of this probe for applications to brain cancer diagnosis in people.

## Discussion

Our results reveal a strategy for functional molecular imaging of physiological processes in the mammalian brain. To detect tumor-associated protease activity, we introduce the FAP-responsive vasoprobe FAPVap, which undergoes a 34-fold increase in activity upon enzymatic cleavage. Using this probe, we demonstrate both spatial localization and pseudo-quantitative characterization of FAP activity over wide fields of view in rodent brains harboring xenografts that display FAP expression comparable to patient-derived tumor cell lines. We also show that the vasoprobe-based contrast mechanism is effective in primates, illustrating the translational applicability of our technology. This study therefore establishes a clinically plausible avenue for brain cancer imaging and provides previously unreported readouts of disease-related enzymatic activity in situ (*Soleimany and Bhatia, 2020*).

The approach presented here exploits the high potency of vasoactive imaging agents and their consequent advantages for noninvasive detection of dilute molecular targets in live subjects (*Desai et al., 2016*; *Ohlendorf et al., 2020*). Notably, vasoprobes like FAPVap are activatable and can be imaged at nanomolar concentrations, even though they do not contain radioactive or magnetic components. Previous efforts to map cancer-associated enzymes in vivo have employed radiotracers for nuclear imaging, superparamagnetic particles for $T_2$-weighted MRI, paramagnetic contrast agents for $T_1$-weighted MRI, and diamagnetic agents for heteronuclear or chemical exchange saturation transfer MRI (*Brindle et al., 2017*; *Hingorani et al., 2015*; *Narunsky et al., 2014*). Among these modalities, the radioactive and superparamagnetic species can both be detected with high sensitivity—at nanomolar or subnanomolar levels—but suffer from other limitations. Radiotracers in particular are not activatable, meaning that they give rise to background signal and must undergo pharmacokinetic partitioning in order to report enzyme activity (*Holland et al., 2013*). Superparamagnetic particles can undergo activation, but their large sizes inhibit delivery to tissue. Paramagnetic and diamagnetic MRI contrast agents are suitable for a number of elegant target-responsive molecular imaging strategies, but high concentrations of these probes are required for most applications (*Wahsner et al., 2019*); this complicates delivery, incurs potential toxicity, and limits prospects for detection of nanomolar-scale targets.

Because FAPVap is far more active as an imaging agent after proteolysis, the time courses of contrast induced by FAPVap provide information about FAP enzyme kinetics in vivo. We used this property to compute a map of FAP-catalyzed product formation rates. Our analysis was based on simplifying assumptions that might require refinement in future studies. For instance, assumptions about probe infusion and washout rates could be solidified by explicitly measuring probe concentration time courses using a CGRP derivative or similar peptide-based probe, perhaps using an alternative image contrast mechanism. The assumption of linearity of the hemodynamic contrast signal could be avoided by incorporating CGRP receptor activation characteristics into the data-fitting procedure

or by using an exogenous vasodilator like $CO_2$ to experimentally define the ceiling of hemodynamic contrast (*Davis et al., 1998*). In vivo specificity of the responses could also be more thoroughly probed; here, we used controls in which FAP was absent, but further controls could utilize inhibitors of FAP or of CGRP-mediated hemodynamic responses. CGRP receptor inhibitors could also be employed to block potential side effects of FAPVap-generated CGRP, which can act as a causal factor in migraine attacks (*Edvinsson, 2019*). Despite the potential for improvements, the measurements we report here in *Figure 3* provide readouts that are stable over independent experiments and demonstrate that the xenograft microenvironment permits both qualitative and quantitative vasoactive molecular imaging in the brain.

In translating FAPVap to larger animal models and eventually to humans, noninvasive brain-wide probe delivery strategies are highly desirable (*Lelyveld et al., 2010*). Here, we used an intra-CSF infusion method that permits relatively low injected doses of the contrast agent to spread over wide fields of view inside the rat brain, but not the periphery. This mode of delivery is similar to clinically applied intrathecal injection methods (*Capozza et al., 2021*; *Fowler et al., 2020*), and with optimization would not require cranial surgery. In the future, even less invasive delivery routes could also be accessible, however, for instance by conjugating FAPVap to blood–brain barrier-permeating carrier proteins such as anti-transferrin receptor antibodies and then injecting the conjugates intravascularly (*Pardridge, 2017*). The low concentrations at which vasoprobes act are likely to make them especially amenable to such approaches. Vasoactive sensors like FAPVap thus offer unique opportunities for spatially comprehensive functional evaluation of molecular targets in brain cancer and other neurological conditions.

## Materials and methods

### Peptide synthesis and conjugation

CGRP-like peptides were made by solid-state synthesis at the MIT Koch Institute Biopolymers lab, oxidatively cyclized, and purified by high-performance liquid chromatography (HPLC). Identity and purity were confirmed by matrix-assisted laser desorption ionization-time-of-flight mass spectrometry and analytical HPLC. After lyophilization, the peptides were weighed, dissolved in water or 50% dimethylsulfoxide, and quantified rigorously using a fluorescent microplate assay (FluoroProfile, Sigma-Aldrich, St. Louis, MO) with CGRP from Sigma-Aldrich as a concentration standard. The peptide solutions were then adjusted to a stock concentration of 100 μM in water and stored at –20°C.

### Proteolytic cleavage reactions

Recombinant human fibroblast activation protein alpha (FAP) expressed in Sf21 cells (400–600 ng/μL) was purchased from Sigma-Aldrich and used at the final concentrations indicated for each experiment. Reactions were performed by combining varying protease concentrations and the desired concentration of the sensor peptide conjugate (10× the highest desired final concentration in the bioassay) in reaction buffer (100 mM NaCl, 20 mM Tris-HCl, pH 7.5), and incubating the mixture at 37°C for 2 hr. These conditions permitted near-complete cleavage, as indicated by our assays. For cleavage reactions followed by serial dilution and CGRP bioassays, 1% 3-[(3-cholamidopropyl) dimethylammonio]–1-propanesulfonate was included in the reaction buffer to reduce adsorptive loss of free peptides. Cleavage reactions performed with titrated amounts of purified FAP or FAP-expressing cells were performed using weights or cell numbers specified in *Figure 4* and incubated under conditions described above, in 100 μL reaction volumes. 10 μL supernatant aliquots were subsequently withdrawn and used in the CGRP bioassay.

### Mammalian cell culture

HEK293FT cells used in this study were purchased from Life Technologies (Grand Island, NY) and U138 MG cells were purchased from ATCC (Manassas, VA). Cells were cultured in 90% DMEM medium, supplemented with 2 mM glutamine, 10% fetal bovine serum (FBS), 100 units/mL penicillin, and 100 μg/mL streptomycin. Cells were frozen in freezing medium composed of 50% unsupplemented DMEM, 40% FBS, and 10% dimethylsulfoxide. The cells tested negative for mycoplasma contamination in the MycoAlert assay (Lonza, Walkersville, MD). For FAP transfection experiments, the HEK293FT cell line was chosen because of its extensive prior use and validation for lentivirus production (*Campeau*

*et al., 2009*) and brain implantation (*Nguyen et al., 2010*). We performed no further authentication of the identity of the cell lines because we obtained them from trusted sources and because functional validation of lentiviral gene transfer was satisfactory.

## Luminescent cAMP assay for CGRP receptor activation

The GloSensor cAMP assay (*Fan et al., 2008*) (Promega) was used to measure cAMP generation upon CGRP receptor activation in real time. This assay system was incorporated into a HEK293FT-based cell line that also expresses both components of the heterodimeric CGRP receptor, forming a reporter cell line that has been described and characterized previously.

To perform assays, 10,000 cells/well were seeded on day 0 in 100 μL DMEM + 10% FBS in white opaque clear-bottom 96-well plates (Costar #3610, Coppell, TX). On day 1 or 2, the medium was removed from the wells and replaced with 90 μL/well of Gibco $CO_2$-independent medium (Life Technologies) + 10% FBS containing 1% v/v of cAMP GloSensor substrate stock solution (Promega). The cells were incubated in substrate-containing medium at 37°C in 5% $CO_2$ for at least 2 hr (maximally 8 hr). Prior to the luminescence bioassay, cells were removed from the cell culture incubator and equilibrated to room temperature and atmospheric $CO_2$ for 30 min. Then, a pre-addition read was performed for 10 min to establish a baseline for luminescence. Compounds and reaction products to be tested were quickly added in 10 μL volume per well at 10× of the desired final concentration using a multichannel pipet. A 30 min time-resolved readout of luminescence was then performed post-addition, with time points acquired every 60 or 90 s, and the time course was examined to confirm a plateau in the signal after 10–15 min, persisting at least through the 25 min time point.

All further data analyses were performed on the basis of the luminescence intensity at the 15 min post-addition time point. Dose–response curves were fitted to a four-parameter Hill equation using Prism (GraphPad Software, San Diego, CA). $EC_{50}$ values are reported as mean and standard deviation of triplicate measurements.

## Plasmids

Lentiviral plasmids were cloned using the Golden Gate method by assembling fragments for the polycistronic expression cassettes into a variant of the pLentiX1 Zeo plasmid with its kanamycin resistance replaced by the ampicillin resistance cassette from pUC18. The lentiviral plasmid used to direct FAP expression in this study contains a gene of interest followed by an internal ribosome entry site (IRES) and a selection marker comprising a fluorescent protein, a 2 A viral sequence (*Szymczak et al., 2004*), and an antibiotic resistance gene. The full sequence of this construct is reported as *Supplementary file 2*. A control plasmid lacking the FAP and IRES components was also generated using otherwise equivalent sequences and cloning methods.

## Lentivirus production and cell line generation

HEK293FT cells were seeded into 6-well plates at 1 million cells/well and transfected using Lipofectamine 2000 (Life Technologies) according to instructions at sub-confluence. Co-transfection of 0.5 μg pMD2.G, 1 μg psPAX2, and 1 μg of the lentiviral plasmid of interest was performed with 6.25 μL Lipofectamine 2000 reagent. Virus-containing supernatant was collected after 48 and 72 hr, filtered through 0.45 μm filters, and used for infection without further concentration. Supernatants were stored at 4°C for up to a week.

HEK293FT cells were seeded into 24-well plates at 40,000 cells/well (final) in the presence of 4 μg/mL polybrene in 50% fresh medium and 50% viral supernatants containing between one and three different viruses. The medium was replaced with fresh viral supernatants daily for 2 days. Selection was performed using both antibiotic resistance and fluorescent markers for each lentivirus. Beginning on day 3 after initial infection, blasticidin at 10 μg/mL (Life Technologies) was added to the medium for selection and selection was continued until all cells expressed fluorescent markers. Selection was then discontinued, cells were expanded, and aliquots were frozen down.

## Preparation of xenograft cells

To prepare cells for implantation into rat brains, HEK293FT cells carrying the recombinant human FAP protein or control lentiviral expression cassettes were seeded at 50% confluence ($7.5 \times 10^6$ cells per plate) in 10 cm cell culture plates and grown to confluence overnight. The next day, the cell layer was

washed twice with fresh medium, aspirated, and scraped. The total volume was adjusted to 150 µL with fresh growth medium for a density of $10^5$ cells/µL and pipetted up and down to break up clumps until a homogenous cell suspension was obtained. Dispersion and integrity of cells was confirmed by brightfield microscopy. FAP expression by transfected cells was assessed using a cleavage assay performed with the fluorogenic substrate H-Ala-Pro-AFC. A standard curve was obtained using purified recombinant FAP concentrations as specified in *Figure 2—figure supplement 1*, and FAP-expressing HEK cells were assayed with approximately $10^5$ cells per 100 µL reaction volume. Supernatants were withdrawn following 1 hr incubation of substrate with samples at 37°C and assayed on a plate reader using 380 nm excitation and 500 nm emission wavelengths.

## Animal procedures

All animal procedures were conducted in accordance with the National Institutes of Health guidelines and with the approval of the MIT Committee on Animal Care. Rodent experiments were performed with male Sprague–Dawley rats, aged 10–12 weeks, supplied by Charles River Laboratories (Wilmington, MA). Nonhuman primate studies were performed with a single adult female squirrel monkey (*Saimiri sciureus*) obtained from the primate colony at MIT.

## Preparation of rats for imaging

Seven Sprague–Dawley rats underwent surgery under isoflurane anesthesia, and bilateral cannula guides (22 gauge, Plastics One, Roanoke, VA) were implanted. For the experiments of *Figure 2* (n = 3), the guides were located over the dorsal striatum, 3 mm lateral to midline, 0.5 mm anterior to bregma, and at a depth of 5 mm from the cortical surface; for the experiments of *Figure 3*, guides were positioned over motor cortex, 2 mm lateral to midline, 3.2 mm anterior to bregma, and at a depth of 1 mm from the cortical surface. For wide-field probe delivery in the experiments of *Figure 3*, an additional cannula guide was implanted on the midline, 5.2 mm anterior to bregma, at a depth of 0.5 mm below the cortical surface. A head post for positioning animals during imaging was attached to each rat's skull using dental cement (C&B Metabond, Parkell, Inc, Edgewood, NY) during the cannula implantation surgery. Cannula guides were sealed with dummy cannulae to avoid exposure of brain tissue during the recovery period. Further experiments were performed after three or more days of postoperative care.

One day before imaging experiments, human FAP-expressing and control HEK293FT cells were xenografted bilaterally through the implanted bilateral cannula guides. Two metal injection cannulae (28 gauge, Plastics One) were attached to 25 µL Hamilton glass syringes and prefilled with the appropriate cell suspensions. Injection cannulae were then lowered into the previously implanted cannula guides. The Hamilton syringes were then placed in a remote infuse/withdraw dual syringe pump (PHD 22/2000; Harvard Apparatus, Holliston, MA), and 3 µL of cell suspension (~0.3 million cells) was injected over 30 min at a rate of 0.1 µL/min. After the cell injection was completed, the metal cannulae were removed and the cannula guides were again sealed with dummy cannulae.

## MRI in rats

Animals were scanned by MRI to measure the changes in hemodynamic contrast following injection of FAPVap probe. Data were acquired on a 7 T scanner with a 20 cm horizontal bore (Bruker Instruments, Ettlingen, Germany). Home-built and commercial (Insight Neuroimaging Systems, Worcester, MA, and Doty Scientific, Columbia, SC) radiofrequency (RF) coils designed for rat brain imaging were used for excitation and detection. During imaging experiments, animals were anesthetized with isoflurane (3% in oxygen for induction; 1% for maintenance). Breathing rate and end-tidal expired isoflurane were continuously monitored. Anesthetized animals were attached via their head posts to a head holder designed to fit within the RF coil systems. Animals with their RF coils were inserted into the magnet bore and locked in a position such that the head of the animal was at the center of the magnet bore.

High-resolution $T_2$-weighted anatomical scans of each animal were obtained using a rapid acquisition with relaxation enhancement (RARE) pulse sequence with effective echo time (*TE*) = 30 ms, repetition time (*TR*) = 5000 ms, RARE factor = 8, matrix size = 256 × 256, 7 coronal slices, and resolution 98 µm × 98 µm × 1 mm (*Figure 2*) or 78 µm × 78 µm × 1 mm (*Figure 3*). Hemodynamic contrast scan series for the experiments of *Figure 2* were obtained also using a RARE pulse sequence with frame time of 5 s, effective TE = 40 ms, TR = 1000 ms, RARE factor = 8, matrix size = 100 × 100, 7

coronal slices, and resolution = 250 µm × 250 µm × 1 mm. Functional imaging scans for *Figure 3* were obtained using gradient echo planar imaging (EPI) with TE = 10 ms, recycle time TR = 4000 ms, matrix size = 64 × 64, 7 coronal slices, and resolution = 390 µm × 390 µm × 1 mm. For each experiment, 5 min of baseline measurements were acquired prior to probe infusion. Following this baseline period, while continuously collecting MRI scans, infusion pumps were remotely turned on to commence infusion of 100 nM FAPVap formulated in aCSF through the bilateral cannulae (injection rate = 0.1 µL/min, *Figure 2* experiments) or midline cannula (injection rate = 1 µL/min, *Figure 3* experiments). Infusion proceeded for 10 min and was followed by further image acquisition.

### $T_1$-weighted contrast agent injections in rats

To assess probe spreading from the intra-CSF infusion procedure described above, two additional rats were implanted with midline cannulae and $T_1$-weighted image series were acquired during injection of Gd-DTPA (Sigma-Aldrich). Each animal was prepared for imaging as described above, except that the intra-CSF infusion syringe was filled instead with 200 mM Gd-DTPA formulated in aCSF. After insertion of animals into the scanner and acquisition of anatomical scans, $T_1$-weighted scans suitable for visualizing Gd-DTPA effects were performed using a fast low-angle shot (FLASH) pulse sequence with TE = 5 ms, TR = 93.8 ms, matrix size = 64 × 64, 7 sagittal slices, and spatial resolution = 400 µm × 400 µm × 1 mm. Scans were collected over a 10 min baseline period followed by 30 min of contrast agent infusion at a rate of 1 µL/min. The resulting data were processed and displayed using MATLAB. Percent signal change values after the injection period were computed with respect to the pre-injection baseline in 1 mm × 1 mm ROIs indicated in *Figure 3B*.

### Analysis of MRI data from rats

MRI data from rats were preprocessed using the AFNI software package (*Cox, 1996*). AFNI routines were applied for motion correction and voxel-wise normalization. Data of *Figure 2* were further subjected to spatial smoothing with a kernel width of 1 mm, and the EPI data of *Figure 3* were aligned to corresponding anatomical scans using the AFNI 3dAllineate function. Additional processing and analysis were performed in MATLAB (MathWorks, Natick, MA). Data were temporally smoothed over a window of 60 s, mean brain signal fluctuations were normalized out, baseline signal averaged over 60 s prior to injection was subtracted from each image series, and signal changes at or outside the edge of the brain were masked out. Resulting data indicated percent signal change with respect to baseline for each voxel within the brain. These were used to compute maps of changes observed after the FAPVap infusion period (*Figures 2A and 3C*, *Figure 3—figure supplement 1*), as well as mean time courses in approximately 1.2 mm × 1.2 mm ROIs placed below the cannula tip locations in each animal. ROI-averaged signal change amplitudes observed at select time points were also calculated. Statistical calculations including *t*-test comparisons between test and baseline or control conditions and *Z*-score calculations were also implemented in MATLAB.

### Kinetic modeling and model fitting

A kinetic model of FAPVap processing is diagrammed in *Figure 2D*. This model was used to simulate the concentrations of the vasoprobe (*V*) and of its FAP-activated product CGRP* (*C*) over time in a voxel or ROI, based on two differential equations:

$$\frac{dV}{dt} = v - (k + v_{out})\,V \tag{1}$$

$$\frac{dC}{dt} = kV - c_{out}C \tag{2}$$

where *k* is the FAP-dependent rate of conversion from *V* to *C*, $v_{in}$ refers to the zeroth-order rate of introduction of the probe, $v_{out}$ refers to the rate of probe efflux, and $c_{out}$ refers to the rate of CGRP* efflux. Note that *k* can be expressed in terms of Michaelis–Menten enzymatic parameters as $k_{cat}$[FAP]/$K_m$, where $k_{cat}$ and $K_m$ are the turnover rate and Michaelis constant, respectively; this equivalence holds where [FAPVap]<< $K_m$, which is justified in our experimental context because local FAPVap concentrations are less than 100 nM and most reported $K_m$ values for FAP lie in the micromolar range (*Schomburg et al., 2017*). For the computed time courses of *Figure 2E*, $v_{in}$ was nonzero only during the indicated infusion period, during which it was set to 0.1 µL/min divided by an ROI volume of

1.56 µL (i.e., 0.064 ROI volumes/min), times the concentration of injected agent, 100 nM, resulting in $v_{in}$ = 0.107 nM/s. The first-order outflow rate $v_{out}$ was modeled as a fixed rate of 0 s$^{-1}$ or 0.01 s$^{-1}$. The FAPVap cleavage rate was estimated based on its equivalence to $k_{cat}$[FAP]/$K_m$, using literature-based values (*Edosada et al., 2006*) of $k_{cat}$ = 3 s$^{-1}$ and $K_m$ = 2 µM for peptide substrates and a FAP concentration of 1 nM or 10 nM, resulting in cleavage rates of $k$ = 0.0015 nM/s or 0.015 nM/s.

For the model-based MRI data analysis of *Figures 2F and 3G*, nonlinear least-squares fitting was used to optimize agreement between observed MRI signal $S$ and modeled signal $\hat{S} = AC(t)$, where $A$ is a constant of proportionality relating the CGRP* concentration to MRI signal change. For the ROI-level fitting of *Figure 2F*, $v_{in}$ was set to 0.107 nM/s, reflecting the ROI volume and infusion rate used in the corresponding experimental analysis. For the voxel-level fitting of *Figure 3G*, $v_{in}$ was crudely estimated by assuming spherically symmetric spread of the agent from the infusion site with constant flux vs. distance, such that

$$v = \frac{r\,[\text{FAPVap}]\,VCS}{4\pi d^2\,VV}$$

(3)

where $r_{in}$ is the infusion rate (1 µL/min), [FAPVap]$_{in}$ is the injected probe concentration (100 nM), $VCS$ is the voxel cross-section (0.15 mm$^2$), $d$ is the distance from the injection site (~3.5 mm), and $VV$ is the voxel volume (0.15 µL), leading to a $v_{in}$ value of 0.0108 nM/s.

For the analysis of *Figure 3G*, the ratio of MRI signal to [CGRP*] represented by the scaling factor $A$ was assumed to be fixed across the entire field of view. $A$ was optimized to a value of 20%/nM using a coarse grid-search over values from 0% to 40%/nM, though fits of roughly similar quality could be obtained over a range of $A$ values. Here, values of $k$, $v_{out}$, and $c_{out}$ were optimized separately for each voxel and the fitting error was computed from all voxels at once. FAP cleavage rates presented in *Figure 3H* reflect fitting to mean data from four animals. Standard errors were computed using jackknife resampling over the four datasets, with individual voxel data points excluded from the SEM calculation when poor fits were achieved ($R^2 < 0.9$) or unphysical enzymatic rates ($k > 10$ s$^{-1}$) were observed. These mean and standard error values were used to perform $t$-tests and voxels with p≤0.05 (n ≥ 2) are identified in the figure. All kinetic modeling and model-fitting was performed using MATLAB.

## Postmortem histology

After MRI experiments, some rats were transcardially perfused with phosphate buffered saline (PBS) followed by 4% formaldehyde in PBS. Brains were extracted, postfixed overnight at 4°C, and then cryoprotected in 30% sucrose for 24–48 hr before sectioning. Free-floating sections (50 µm) were cut using a vibratome (Leica VT1200 S, Leica Microsystems GmbH, Wetzlar, Germany), mounted on glass slides with VECTASHIELD mounting medium with DAPI (Vector Laboratories, Burlingame, CA) and protected with a coverslip. The distribution of injected HEK293FT cells was indicated by red fluorescence due to mKate2 reporter expression in both FAP-expressing and control cells.

## Preparation of squirrel monkey for optical imaging

Squirrel monkeys were pair-housed in an environment controlled for temperature (20–23°C), humidity (30–70%), and photoperiod (12 hr dark/12 hr light). Their cages were equipped with a variety of perches and enrichment devices, and they received regular health checks and behavioral assessment from attending veterinarians and researchers.

For in vivo optical imaging during application of vasoprobe agents, a resealable chamber with an optically clear glass window and ports for drug application was custom-designed and 3D-printed (*Figure 4—figure supplement 1*). The circular polycarbonate chamber had a central opening of 5 mm diameter over which a clear glass coverslip (Werner Instruments, Holliston, MA) was placed for optical imaging. Two pairs of silicone O-rings sandwiched the glass coverslip, and a finely threaded polycarbonate O-ring was used to tighten the assembly and prevent any leakage. The chamber was equipped with an inlet and an outlet port, at opposite ends, for controlled fluid circulation at a specified rate via a syringe infusion pump.

In preparation for the procedure to surgically implant the optical imaging chamber, a single monkey was placed on food restriction overnight. On the day of the surgery, the animal was premedicated with atropine (0.04 mg/kg) and sedated with ketamine (30 mg/kg), both given intramuscularly.

Following intubation, anesthesia was induced with 3% sevoflurane and later maintained at 1.5% sevoflurane mixed with $O_2$. Sensors and electrodes for continuous recording of heart rate, oxygen saturation, electrocardiograms, end-tidal $CO_2$, and rectal temperature were attached and intravenous (IV) catheter was placed in the saphenous vein for infusion of fluids and drugs during the surgery and recovery period. Once the animal acquired a stable plane of anesthesia, it was placed in a stereotaxic apparatus. A thin layer of sterile eye lubricant was applied to protect against corneal drying. A single bolus dose of intravenous dexamethasone (0.4 mg/kg) was provided to guard against brain swelling. The scalp and fascia were removed in layers via blunt dissection to expose the implant site, and craniotomy was performed with a diameter large enough to tightly fit the lower pedestal of the optical chamber, which was then cemented to the skull with C&B Metabond. A small cap covering the optical chamber was held in place with four skull screws custom-designed in-house from MRI-compatible, medical grade PEEK. The fascia and scalp were then sutured around the implant.

After surgery, the animal was provided with anti-inflammatory pain relief medications and antibiotics as prescribed by the attending veterinary staff. The animal was then allowed to recover and extubated once fully conscious. This process took place in a recovery cage until the monkey was able to move and sit unaided; it was then returned to its home cage. The animal was closely monitored for the next several hours and then several times daily over the next 72 hr.

## Squirrel monkey optical imaging and analysis

After a period of 3 weeks, allowing for full recovery from cephalic implant procedure, optical imaging experiments began. Following standard food restriction and premedication protocols, the animal was intubated and anesthetized with 3% sevoflurane and placed in a stereotaxic frame. In preparation for imaging, the area around the optical chamber was cleaned and disinfected with betadine and alcohol scrubs, and a Steri-Drape (3M, St. Paul, MN) with an opening to expose the imaging chamber was placed over the monkey's head. The optical chamber cover was removed, and the area was flushed with sterile prewarmed saline to remove any tissue and debris. The optical window was replaced by fresh sterile glass coverslip and sealed tightly after carefully removing any air bubbles. The inlet and outlet ports were then used for drug application and irrigation with sterile aCSF.

A charge-coupled device camera (Prosilica GC, Allied Vision, Newburyport, MA) attached to a dissection microscope (Stemi SV11 M2 Bio, Carl Zeiss AG, Oberkochen, Germany) was used to image the cortical surface at a frame rate of approximately 4 Hz. Illumination was regulated using a xenon arc lamp. A green band-pass filter (550 ± 25 nm) was used to provide optimal vascular contrast. A 100 nM CGRP (human α-CGRP, Sigma-Aldrich) solution was prepared in aCSF. To establish imaging baselines, aCSF alone was continuously infused (30 µL/min) through the inlet port for 5 min prior to CGRP infusion; images were acquired continuously throughout. Fluid was extracted from the imaging chamber through the outlet port at an equal rate in order to maintain a constant pressure. At the end of the baseline period, the CGRP solution was infused at a rate of 30 µL/min.

Data were temporally down-sampled to 1 Hz and processed further in MATLAB. Because of a fluid-level artifact upon fluid switching, image signal changes were analyzed starting at CGRP infusion onset (defined as $t = 0$). This included measurement of vessel diameter by examining the full width at half-height of the vessel transected by the red line in *Figure 4A*, as well as measurement of cortical reflectance from the parenchymal region identified by the black box in this figure panel. Values are expressed as percent change from their $t = 0$ values.

## Preparation of squirrel monkey for MRI

In preparation for MRI experiments, the same monkey used for optical imaging was transferred to a custom-built anesthesia induction box. Anesthesia was induced with sevoflurane (4% in $O_2$) for ~5 min. After the animal was immobile, suppression of reflexes was verified, and it was taken out of the chamber and intubated using a silicone-cuffed 2.5 mm inner diameter endotracheal tube (ET) without wire reinforcement (Med-Caire, Vernon, CT). The tube was fitted to span the distance from the mouth to the manubrium sternum. Prior to intubation, a single spray of the topical anesthetic Cetacaine (Cetylite Industries, Pennsauken, NJ) was delivered to the glottis to reduce the incidence of laryngospasm. The ET was coated with lidocaine hydrochloride oral topical solution and inserted into the trachea using a stylet and laryngoscope (size 1 Macintosh blade). Successful ET placement was determined by observing motion of hairs held at the opening of the ET connector, condensation on a

mirror, and expansion/contraction of a latex covering placed at the end of the ET connector. The cuff on the ET was then inflated with air. The ET was secured with a velcro strap customized with a 15 mm opening that fit around the ET connector and wrapped around the head.

Anesthesia was reduced to 2.5% sevoflurane and an IV catheter was placed in the saphenous vein. The catheter was a 24G × 3/4-inch Surflo model with a 27G needle and Surflo injection plug (Terumo Medical Corp., Somerset, NJ). The IV line was then used to deliver lactated Ringer's solution (7.5 mL/kg/hr). A one-time dose of anticholinergic glycopyrrolate (0.004 mg/kg IM) was provided at the same time. The animal was placed on mechanical ventilation (SAR-830 Series Small Animal Ventilator, CWE, Inc) and anesthesia was further reduced to 2% sevoflurane in oxygen (0.5–0.6 L $O_2$/min), while the animal was positioned in a custom cradle incorporating ear bars and head restraint.

In preparation for infusion of vasoprobe and control solutions during MRI, the previously implanted chamber for optical imaging was gently opened by removing the glass cover slip. A 22G Teflon cannula was inserted in the cortex in the radial direction to a depth of 6 mm below cortical surface. The cannula was connected to the infusion pump via polyvinyl tubing (3 mm internal diameter). A similar cannula was placed in a different location, 5 mm away and at a depth of 10 mm, for infusion of aCSF control solution via a small craniotomy. Sevoflurane was then reduced in three incremental steps over 30 min to achieve 0.6% (in balance of oxygen) expired sevoflurane level, measured with a SurgiVet V9004 Capnograph (Smiths Medical, St. Paul, MN). Ventilation was maintained at a rate between 34 and 39 breaths per minute with an inspiration time of 0.5 s and expiration duration of 1.1 s. Breathing rate and end-tidal anesthetic were continuously monitored throughout subsequent imaging, and animal core temperature was maintained at approximately 37°C using a heating blanket. Following MRI experiments, a 3 mL injection of 5% dextrose was delivered after the animal was removed from the scanner.

## Squirrel monkey MRI acquisition and analysis

MRI of the squirrel monkey was performed using a Bruker Biospec 9.4 T scanner with 20 cm bore, equipped with a custom-built surface transceiver coil. A high-resolution $T_2$-weighted anatomical scan was obtained using a RARE pulse sequence with effective TE = 35.5 ms, TR = 2500 ms, RARE factor = 8, spatial resolution = 156 μm × 156 μm × 1 mm, and matrix size = 256 × 256 with 10 coronal slices around the injection site. Hemodynamic contrast image series were acquired using a gradient EPI pulse sequence with TE = 14 ms, TR = 8000 ms, spatial resolution = 400 μm × 400 μm × 1 mm, and matrix size = 100 × 100 with 10 slices. Five minutes of baseline measurements were acquired prior to probe infusion. Following this baseline period, while continuously collecting EPI images, infusion pumps were remotely turned on to commence intracranial injection of 100 nM CGRP or control aCSF solutions at a rate of 0.1 μL/min for 10 min through the preimplanted cannulae.

Functional imaging data were motion corrected in AFNI and imported into MATLAB for further analysis. Each voxel's image intensity was temporally smoothed using a running window of 40 time points, normalized to the preinfusion baseline, and converted to units of percent signal change. A map of percent signal change in brain voxels was computed for the 20 min time point and superimposed over the anatomical scan. 1.6 mm × 1.6 mm ROIs were specified at the cannula tips used for CGRP and aCSF injections and also at two arbitrarily placed control locations in the same brain slice. Mean percent signal change amplitudes in each ROI were plotted versus time.

## Acknowledgements

This research was supported by NIH grants R24 MH109081, U01 NS103470, and U01 EB031641 and by grants from the MIT Simons Center for the Social Brain and the G. Harold and Leila Y. Mathers Foundation to AJ. ALS was supported by predoctoral fellowships from the Boehringer-Ingelheim Fonds and the Friends of the McGovern Institute. AC was supported by the SCSB undergraduate research opportunities program. RO was funded by a fellowship from the Deutsche Forschungsgemeinschaft. AW was funded by the Advanced Multimodal Neuroimaging Training Program at the Massachusetts General Hospital (R90 DA023427) and a fellowship from Harvard-MIT Health Sciences and Technology program.

## Additional information

### Funding

| Funder | Grant reference number | Author |
|---|---|---|
| National Institute of Mental Health | R24 MH109081 | Alan Jasanoff |
| National Institute of Neurological Disorders and Stroke | U01 NS103470 | Alan Jasanoff |
| National Institute of Biomedical Imaging and Bioengineering | U01 EB031641 | Alan Jasanoff |
| MIT Simons Center for the Social Brain | | Alan Jasanoff |
| G. Harold and Leila Y. Mathers Foundation | | Alan Jasanoff |

The funders had no role in study design, data collection and interpretation, or the decision to submit the work for publication.

### Author contributions

Mitul Desai, Conceptualization, Data curation, Formal analysis, Investigation, Methodology, Visualization, Writing - original draft, Writing – review and editing; Jitendra Sharma, Data curation, Investigation, Methodology, Visualization, Writing – review and editing; Adrian L Slusarczyk, Conceptualization, Data curation, Investigation, Methodology, Writing – review and editing; Ashley A Chapin, Conceptualization, Writing – review and editing; Robert Ohlendorf, Formal analysis, Visualization, Writing – review and editing; Agata Wisniowska, Data curation, Investigation, Methodology, Writing – review and editing; Mriganka Sur, Resources, Writing – review and editing; Alan Jasanoff, Conceptualization, Data curation, Formal analysis, Funding acquisition, Methodology, Project administration, Resources, Supervision, Visualization, Writing - original draft, Writing – review and editing

### Author ORCIDs

Robert Ohlendorf ⓘ http://orcid.org/0000-0002-1720-4400
Mriganka Sur ⓘ http://orcid.org/0000-0003-2442-5671
Alan Jasanoff ⓘ http://orcid.org/0000-0002-2834-6359

### Ethics

All animal procedures were conducted in accordance with National Institutes of Health guidelines and with the approval of the MIT Committee on Animal Care. Rodent procedures were approved as part of protocol 0718-068-21. Squirrel monkey procedures were approved as part of protocol 0120-001-23.

### Decision letter and Author response

Decision letter https://doi.org/10.7554/eLife.70237.sa1
Author response https://doi.org/10.7554/eLife.70237.sa2

## Additional files

### Supplementary files

• Supplementary file 1. Sequences of candidate fibroblast activation protein (FAP) probes.
• Supplementary file 2. Fibroblast activation protein (FAP) expression plasmid sequence.
• Transparent reporting form

### Data availability

MRI scan series analyzed in this study have been deposited in the Dryad database (doi:https://doi.org/10.5061/dryad.31zcrjdkp).

The following dataset was generated:

| Author(s) | Year | Dataset title | Dataset URL | Database and Identifier |
|---|---|---|---|---|
| Jasanoff A | 2021 | Data from: Hemodynamic molecular imaging of tumor-associated enzyme activity in the living brain | http://dx.doi.org/10.5061/dryad.31zcrjdkp | Dryad Digital Repository, 10.5061/dryad.31zcrjdkp |

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
