## [Editor Report]

Molecular probes that respond to disease-specific activities to produce a diagnostic readout have had a major impact in the clinical management of cancer. The current study describes a genetically engineered sensor for the cancer-associated, fibroblast activation protein, which reports via local changes in hemodynamic image contrast using MRI. Development of activity based imaging probes is an important area of study for advancing precision medicine and may more accurately represent disease prognosis and stratification over conventional imaging probes.

---

## [Decision Letter]

**Decision letter after peer review:**

Thank you for submitting your article "Hemodynamic molecular imaging of tumor-associated enzyme activity in the living brain" for consideration by *eLife*. Your article has been reviewed by 2 peer reviewers, one of whom is a member of our Board of Reviewing Editors, and the evaluation has been overseen by Wafik El-Deiry as the Senior Editor. The reviewers have opted to remain anonymous.

Essential revisions:

1) in vivo experiments presented utilize HEK293 cells and not GBM tumor cells. In addition, FAP expression is driven by an exogenous transgene. Besides the practicality of this approach, the biological significance of the experimental design is a weakness since the claim is made that the FAP sensor can detect "tumor associated enzyme activity". A careful survey of patient derived samples should be conducted to provide confidence that FAPVap can detect a majority of GBM. In addition, a FAP positive tumor model should be investigated.

2) There is an analysis of FAP catalysis rate, k, based on time course imaging in Figure 3H, however a range of catalysis rates were observed. From this reviewer's knowledge, for a known substrate and protease, the catalysis rate would be expected to be constant. An explanation of the observations made in Figure 3H would be useful.

3) While the total amount of FAP per cell was measured, it would be useful if it was put in the context of physiologically-relevant values.

*Reviewer #1 (Recommendations for the authors):*

Critique:

The described findings are an extension of their published work that first described the FAP sensor. In the current paper, the FAP sensor is optimized and used in rodent models to demonstrate that the sensor activation can be detected through local changes in hemodynamic image contrast using T2 and T2* MRI. The sensitivity and signal to noise of the system is impressive and a major strength of the described research.

Weaknesses that should be addressed include:

(1) in vivo experiments presented utilize HEK293 cells and not GBM tumor cells. In addition, FAP expression is driven by an exogenous transgene. Besides the practicality of this approach, the biological significance of the experimental design is a weakness since the claim is made that the FAP sensor can detect "tumor associated enzyme activity".

(2) CGRP blockers have been introduced or are in late-stage trials for migraine. Specificity of the sensor could be demonstrated using these reagents.

(3) The manuscript states that the described findings establish FAPVap as a clinically feasible avenue for brain cancer imaging. It is not obvious if FAP is consistently expressed in GBM, if not, the statement is premature. A careful survey of patient derived samples should be conducted to provide confidence that FAPVap can detect a majority of GBM.

(4) Several lines of evidence implicate CGRP as a key effector molecule in the migraine attack. The safety of the FAPVap should be discussed since the released CGRP in FAP positive tissues may lead to unintended pathophysiology.

(5) The novelty of the described studies is in the finding that MRI can be used to detect FAP catalytic activity using MRI. This is impressive, as are the in vitro studies that quantitatively evaluate the sensitivity of the FAPVap sensor. However, in vivo titration experiments would be informative to get a better understanding of the sensitivity of the FAPVap when MRI is used as a readout in the brain.

*Reviewer #2 (Recommendations for the authors):*

The flow of Figure 1 would be easier to understand if panel D appeared before panel C.

There is an analysis of FAP catalysis rate, k, based on time course imaging in Figure 3H, however a range of catalysis rates were observed. From this reviewer's knowledge, for a known substrate and protease, the catalysis rate would be expected to be constant. An explanation of the observations made in Figure 3H would be useful.

Please include c_out_ as a parameter in the model in Figure 2D. It is very difficult to distinguish the difference between dotted and dashed lines in Figure 2E – consider changing the presentation.

In Figure 3D, why does the noise increase in the control measurement over time?

It would be useful if the plots in Figure 3G were mapped to the specific voxels they represent.

While the total amount of FAP per cell was measured, it would be useful if it was put in the context of physiologically-relevant values.

---

## [Author Response]

Essential revisions:1) In vivo experiments presented utilize HEK293 cells and not GBM tumor cells. In addition, FAP expression is driven by an exogenous transgene. Besides the practicality of this approach, the biological significance of the experimental design is a weakness since the claim is made that the FAP sensor can detect "tumor associated enzyme activity". A careful survey of patient derived samples should be conducted to provide confidence that FAPVap can detect a majority of GBM. In addition, a FAP positive tumor model should be investigated.

We provide new data from a FAP positive human glioma cell line part of an updated Figure 4 in the revised manuscript. Our results show that U138 cells activate the FAPVap probe to an extent similar to the FAP-expressing cell line we used for our in vivo experiments. Both cell types display FAP activity at levels corresponding to approximately 1 pg FAP/cell. We have also surveyed a range of patient-derived glioma cell lines discussed in the literature; the cells we studied display FAP activity in an intermediate range, with a majority of glioma lines surpassing U138 cells in activity, while some cell lines display lower FAP levels. This further supports plausibility of our approach for clinically relevant imaging of FAP activity in brain cancer. Our new data is presented in a revised version of Figure 4, with corresponding results text on p. 9, as well as further textual adjustments elsewhere in the manuscript.

2) There is an analysis of FAP catalysis rate, k, based on time course imaging in Figure 3H, however a range of catalysis rates were observed. From this reviewer's knowledge, for a known substrate and protease, the catalysis rate would be expected to be constant. An explanation of the observations made in Figure 3H would be useful.

We have added new text to better explain the biochemical meaning of *k*, which functions as an effective first order rate constant for FAP-mediated product formation from FAPVap. In this regard, *k* differs from the *k_cat_* defined in standard Michaelis-Menten nomenclature, which is indeed a constant independent of enzyme concentration. Instead, the product formation rate constant we utilize is approximately equal to *k_cat_*[FAP]/*K_m_* in the Michaelis-Menten formalism, where *k_cat_* and *K_m_* are the turnover rate and Michaelis constants and [FAP] is the local enzyme concentration. This equivalence arises from the fact that FAPVap concentrations in our experiments (< 100 nM) are much lower than reported FAP *K_m_* values for similar substrates (> 10 µM). Revised text clarifying these points has been added on p. 6 and 21. In addition, the kinetic model incorporating *k* is fully described by Equations 1-2 on p. 21 and diagrammed in Figure 2D. On pp. 8-9 of the revised paper, we further clarify with regard to Figure 3H that variation in *k* values is expected to arise primarily from variations in FAP concentration.

3) While the total amount of FAP per cell was measured, it would be useful if it was put in the context of physiologically-relevant values.

The revised paper includes a direct comparison of enzymatic activity levels measured using FAPVap from purified FAP, FAP-expressing HEK cells, and the U138 human glioma cell line (new Figure 4A,B). Levels of FAP activity on both recombinant and tumor-derived cells are equivalent to roughly 1 pg FAP/cell. As also noted above, FAP expression by these cells is comparable to reported FAP expression on a number glioma cell lines, indicating its physiological relevance (see relevant text on p. 9).

Reviewer #1 (Recommendations for the authors):Critique:The described findings are an extension of their published work that first described the FAP sensor. In the current paper, the FAP sensor is optimized and used in rodent models to demonstrate that the sensor activation can be detected through local changes in hemodynamic image contrast using T2 and T2* MRI. The sensitivity and signal to noise of the system is impressive and a major strength of the described research.Weaknesses that should be addressed include:(1) In vivo experiments presented utilize HEK293 cells and not GBM tumor cells. In addition, FAP expression is driven by an exogenous transgene. Besides the practicality of this approach, the biological significance of the experimental design is a weakness since the claim is made that the FAP sensor can detect "tumor associated enzyme activity".

We provide new FAPVap sensitivity data from a FAP positive human glioblastoma cell line in the revised manuscript. Our results show that U138 tumor cells activate the FAPVap probe to an extent similar to the FAP-expressing cell line we used for our in vivo experiments. Both cell types display FAP activity at levels corresponding to approximately 1 pg FAP/cell. The new data is presented in a revised version of Figure 4, with corresponding results text on p. 9, as well as further textual adjustments elsewhere in the manuscript.

(2) CGRP blockers have been introduced or are in late-stage trials for migraine. Specificity of the sensor could be demonstrated using these reagents.

An advantage of performing the xenograft comparison between control HEK cells and FAPexpressing HEK cells is that factors other than FAP are conserved between the conditions and therefore controlled for. It is true nevertheless that both FAP inhibitors and hemodynamic blockers including CGRP receptor inhibitors could be used to perform additional control experiments. We note the possibility of doing this in revised text on p. 12.

(3) The manuscript states that the described findings establish FAPVap as a clinically feasible avenue for brain cancer imaging. It is not obvious if FAP is consistently expressed in GBM, if not, the statement is premature. A careful survey of patient derived samples should be conducted to provide confidence that FAPVap can detect a majority of GBM.

We have surveyed a range of patient-derived glioma cell lines discussed in the literature and find that the U138 cells we now analyze in Figure 4 display FAP activity in an intermediate range, with a majority of glioma lines surpassing these cells in activity, while some cell lines display lower FAP levels. This further supports plausibility of our approach for clinically relevant imaging of FAP activity in brain cancer. At the same time, we have made edits at multiple points in the paper to avoid overstating the case for clinical feasibility at this point.

(4) Several lines of evidence implicate CGRP as a key effector molecule in the migraine attack. The safety of the FAPVap should be discussed since the released CGRP in FAP positive tissues may lead to unintended pathophysiology.

We now note the role of CGRP in migraine on p. 12 of the revised text, along with the possibility of using CGRP inhibitors to block FAPVap side effects if required for this reason.

(5) The novelty of the described studies is in the finding that MRI can be used to detect FAP catalytic activity using MRI. This is impressive, as are the in vitro studies that quantitatively evaluate the sensitivity of the FAPVap sensor. However, in vivo titration experiments would be informative to get a better understanding of the sensitivity of the FAPVap when MRI is used as a readout in the brain.

In the revised paper, we have added a sensitivity analysis that utilizes data from the experiments of Figure 4, which cover a range of effective FAP levels observed over the ROIs we investigated. The analysis, shown in the new Figure 3—figure supplement 2 and discussed on pp. 8-9, indicates that FAP concentrations of about 13 nM or more could be detectable in vivo at a single voxel level in individual animals according to the methods and data of Figure 4. What this concentration corresponds to in terms of number of FAP-expressing cells depends on the spatial distribution and FAP activity levels exhibited by the cells in question, but our in vitro data suggest that a density of 10^3^ FAP-positive cells/µL displays activity corresponding to ~10 nM FAP (Figure 2—figure supplement 1).

Reviewer #2 (Recommendations for the authors):The flow of Figure 1 would be easier to understand if panel D appeared before panel C.

Data in the style of panel 1C is the basis for determining EC_50_ values as presented in panel 1D, so we consider it more natural to keep the order as is. We introduce a new sentence of text on p. 4, however, to explain the logic of this choice better.

There is an analysis of FAP catalysis rate, k, based on time course imaging in Figure 3H, however a range of catalysis rates were observed. From this reviewer's knowledge, for a known substrate and protease, the catalysis rate would be expected to be constant. An explanation of the observations made in Figure 3H would be useful.

We have added new text to better explain the biochemical meaning of *k*, which functions as an effective first order rate constant for FAP-mediated product formation from FAPVap. In this regard, *k* differs from the *k_cat_* defined in standard Michaelis-Menten nomenclature, which is indeed a constant independent of enzyme concentration. Instead, the product formation rate constant we utilize is approximately equal to *k_cat_*[FAP]/*K_m_* in the Michaelis-Menten formalism, where *k_cat_* and *K_m_* are the turnover rate and Michaelis constants and [FAP] is the local enzyme concentration. This equivalence arises from the fact that FAPVap concentrations in our experiments (< 100 nM) are much lower than reported FAP *K_m_* values for similar substrates (> 10 µM). Revised text clarifying these points has been added on p. 6 and 21. In addition, the kinetic model incorporating *k* is fully described by Equations 1-2 (p. 21) and diagrammed in Figure 2D. On pp. 8-9 of the revised paper, we further clarify with regard to Figure 3H that variation in *k* values is expected to arise primarily from differences in FAP local concentrations.

Please include c_out_ as a parameter in the model in Figure 2D. It is very difficult to distinguish the difference between dotted and dashed lines in Figure 2E – consider changing the presentation.In Figure 3D, why does the noise increase in the control measurement over time?

We have added *c_out_* to the model of Figure 2D, as advised. The dashed lines in Figure 2E have been reformatted to make them more distinct from the dotted lines. The apparent noise in Figure 3D increases because of signal drift trends that differ among subjects and thus increase the standard error of the mean value over time.

It would be useful if the plots in Figure 3G were mapped to the specific voxels they represent.While the total amount of FAP per cell was measured, it would be useful if it was put in the context of physiologically-relevant values.

The plots in Figure 3G represent voxels evenly spaced over the white box in Figure 3F; this is now noted in the caption. The new data in Figure 4A compare FAP activity levels between the FAP-expressing HEK cells we used for in vivo experiments and the human-derived glioma cell line U138. The levels are the same and according to the standard curve of Figure 4B correspond to roughly 1 pg FAP/cell. In addition to these new data, we include a new sensitivity analysis of the data in Figure 3. The new analysis, presented in Figure 3—figure supplement 2 and discussed on pp. 8-9, characterizes the concentration of FAP we would expect to be able to detect at a single voxel and single animal level using current methods. The resulting value of ~10 nM corresponds to a FAP-positive cell density of about 10^3^ cells/µL, according to data of Figure 2—figure supplement 1.